# Deception is associated with reduced social connection

Samantha Sprigings [1], Cameo J. V. Brown [1] & Leanne ten Brinke [1✉]

Lies can have major consequences if undetected. Research to date has focused primarily on the consequences of deception for receivers once lies are discovered. We advance deception research and relationship science by studying the social consequences of deception for the sender—even if their lies remain undetected. In a correlational study of video conversations (Study 1; $N = 776$), an experimental study of text conversations (Study 2; $N = 416$), and a survey of dispositional tendencies (Study 3; $N = 399$), we find consistent evidence that people who lie tend to assume that others are lying too, and this impedes their ability to form social connections. The findings provide insight into how (dis)honesty and loneliness may go together, and suggest that lies—even when undetected—harm our relationships.

[1] Department of Psychology, University of British Columbia Okanagan, Kelowna, BC, Canada. ✉email: leanne.tenbrinke@ubc.ca

Social connection is a fundamental human motive[1]. Loneliness, or the subjective sense of lacking social connection, is associated with depression, poor sleep quality, excessive stress reactivity, decreased immunity, and increased mortality[2]. It is therefore critical to understand how to foster social connection. It is clear that social connections can be forged through conversation; particularly those characterized by reciprocal disclosure —where both parties share personal information[3,4]. Sharing such information requires trust, as it creates vulnerability between the parties[5]. However, conversations also provide a stage for trust to be violated. For example, people admit to telling an average of one or two lies per day, with a minority of individuals engaging in many more[6,7]. Although people generally tell the truth and admit to lying only rarely, discovering these falsehoods can damage trust between parties[8]. Depending on the nature of the lie, discovering deception can even result in relationship dissolution[9]. We expand on this work by positing that an erosion of trust and social connection occurs immediately upon telling a lie, and even if lies are never unveiled. Rather than focusing on the receiver's reaction to deception, we focus on the social consequences of deception for the sender. We hypothesize that telling lies contributes to a diminished sense of social closeness, or loneliness, for the sender by diminishing their trust in others.

Deception is the act of intentionally misleading another, and may include complete falsifications (i.e., lies), distorting, or concealing the truth[10,11]. Engaging in deception differs from truth-telling on many dimensions. For example, deception is more cognitively taxing than telling the truth[10,12]. Whereas the truth can be relayed from memory, deceptive statements must be invented to be plausible and avoid contradicting prior knowledge of the receiver. Additionally, deception may result in greater (or different) emotional arousal relative to truth-telling[13]. A liar may feel guilty about their behavior, or nervous about their lies being discovered. Alternatively, deception may involve attempting to falsify emotions that are not actually felt, or hiding experienced emotions[14]. Other research suggests that deception may be associated with more impression management, relative to telling the truth. Attempts to appear honest might backfire if a liar overcontrols their behavior, resulting in an unnaturally stiff impression[15,16]. These theories are purported to predict behavioral cues that may reveal deception in real-time[11]. Indeed, a primary focus of research on deception has been on detecting its behavioral manifestations, with relatively little attention on the internal experience and intrapersonal consequences of this common conversational act.

People sometimes engage in deception because they are seeking personal gain or to avoid negative consequences[17]. Other times, people avoid having honest conversations because they believe that it will be uncomfortable, unpleasant, or perceived as unkind. However, recent research suggests that these concerns may be misplaced. When randomly assigned to spend three days being absolutely honest or kind, participants overestimated the social connection associated with kindness and underestimated the social connection associated with honesty[18]. Although this research did not include a lie condition, it suggests that engaging in deception may have negative consequences for feelings of social connection. Relatedly, research has examined the process and consequences of keeping secrets, which are conceptually similar to deception in that they also involve concealing the truth[19]. People report that they feel 'alone' with their secrets and that keeping secrets makes them feel isolated from others[20]. Similarly, a longitudinal study found that adolescent secret-keeping was bi-directionally associated with decreased parent-child relationship quality[21]. Thus, while research has not examined the effect of telling lies on social connection in dyadic conversations with strangers or on feelings of loneliness, related work supports this possibility.

While it might seem obvious that receiving a lie (and discovering it) would lead to an erosion of trust and social connection, it is less clear why telling a lie would lead to the same outcome. Ironically, we argue that lies arouse distrust in the sender—making them less trusting of their (honest) conversational partners. Specifically, a false consensus of distrust is expected to explain the relationship between sender veracity and feelings of social connection. That is, people tend to use their own behavior to infer how others would act in the same situation[22]. Relatedly, because people generally operate on a truth bias and rarely consider the possibility of deception in most interactions[23,24], one's own deception may act as a trigger that raises suspicion about trust violations. Consistent with this explanation, a phenomenon known as deceiver's distrust has been described, finding that senders who tell lies (vs. truths) perceive receivers to be less honest[25]. Similarly, there is support for a false consensus bias concerning deception in the discovery phase of dating[26]. Participants' self-reported dishonesty was positively related to their perceptions of others' dishonesty. For instance, adolescents (8 to 17 years old) who cheated and lied about peeking at answers on a test were biased towards believing that their peers would have done the same[27]. In short, there is considerable research to suggest that there is a false consensus concerning deception (i.e., deceiver's distrust). Accordingly, liars may be less likely to trust their conversational partners than truth-tellers as a reflection of their own behavior, rather than any actual dishonesty by their partner, thereby diminishing the foundation of trust necessary to forge social connection.

In three studies, we examine how trustworthy communication and honesty support social connection and, inversely, how deception can doom relationships from the outset. Specifically, we predict that engaging in deception will be associated with decreased social connection [H1] by decreasing trust in others [H2] in free-flowing dyadic conversations (Study 1) and in tightly-controlled dyadic conversations where sender veracity is manipulated (Study 2). We also expect that people's general dispositions for telling lies, trusting others, and feeling lonely will reflect a similar pattern such that people who tell more lies will also report trusting others less and will feel greater loneliness in their lives (Study 3).

## Methods
Our studies were not pre-registered. In our initial study, we leverage an existing conversational database to test whether there is a positive relationship between self-ratings of trustworthiness and reported feelings of closeness with a conversational partner [H1]. Additionally, we test whether perceived trustworthiness of one's partner mediates this relationship [H2].

**Study 1 Corpus**. BetterUp Inc. released a multimodal dataset of naturalistic conversations collectively referred to as the CANDOR corpus (Conversation: A Naturalistic Dataset of Online Recordings)[28]. The corpus includes over 1TB of data, including raw and processed recordings, transcripts, behavioral measures, and survey responses from a large, diverse sample of participants based in the United States. This study received approval by Ethical & Independent Review Services.

Participants were recruited using Prolific; recruitment targeted individuals based in the United States and 18 years of age or older. Between January and November 2020, six rounds of data collection yielded a total of 1656 dyadic conversations that were recorded over video chat. Participants provided informed consent to have a conversation with a stranger that would last at least 25 minutes, complete survey ratings of their experience, and have their data made publicly available. Dyads included in our analyses

conversed for 29 to 113 minutes with an average conversation length of 30.01 minutes ($SD = 7.79$). Participants were paid $0.85 for completing an initial scheduling survey and an additional $14.15 upon full completion of the recorded conversation and post-conversation survey.

We received the CANDOR corpus in March 2022. Dyads were included in analyses if both participants completed the post-conversation ratings of personal trustworthiness, perceived trustworthiness of their partner, and the measure of interpersonal closeness with their partner. No further data exclusions were made. This resulted in a total of 388 complete dyads in the analyses below. Of these 776 participants, 423 identified as female, 321 as male, and 5 as other or prefer not to answer. Participants were an average age of 33.81 ($SD = 10.98$; range = 19–63). Twenty-seven participants did not provide age information.

**Study 1 Procedures**. Dyads were matched according to their shared availability, which participants reported in an initial survey. Once matched, participants were notified via email of the time and date of their conversation.

A brief survey was conducted to measure participants' current mood prior to their conversation[28]. This survey instructed participants to make sure their webcam and microphone were enabled and to have a conversation lasting at least 25 minutes. Then, a link was provided to the video chat room, where the conversation was to take place.

As soon as the first participant clicked the link from the pre-conversation survey, the recording began. Participants were not given any specific instructions regarding conversation content. Instead, they were told to "talk about whatever you like, just imagine you have met someone at a social event and you're getting to know each other." After completing the conversation, participants ended the recording session and returned to their original web browser to complete the post-conversation survey.

Following the conversation, participants reported on their experience of the conversation and perceptions of their conversation partner's and their own psychological states and traits[28]. Importantly for the purposes of this investigation, participants responded to the statements, "How would you rate yourself on each of the following traits?—trustworthy?", and "To what extent does your conversation partner have each of the following traits?—trustworthy?", on a 1 (not at all) to 9 (extremely) scale. Additionally, participants were asked to rate their agreement with the statement "I felt close to my partner" on a 1 (strongly disagree) to 7 (strongly agree) scale. Following completion of the post-conversation survey, participants were thanked for their time and provided with information on how to make a request for payment.

A limitation of relying on the CANDOR dataset[28] to test our hypotheses is that participants responded to a single item, trait-measure of their personal trustworthiness, but did not provide details on the trustworthiness of their communication in the conversation, specifically. In Study 2, we manipulated sender veracity in the context of a dyadic conversation and subsequently measured feelings of closeness with the conversational partner, allowing us to establish a causal link between telling lies and feeling socially distant [H1]. We also examine 'deceiver's distrust' as a potential mechanism for this relationship [H2].

**Study 2 Participants**. Our sample size goal ($N = 200$ dyads) was identified on the basis of similar research, involving stranger dyads in a chat-based, two-group experimental design[29]. Participants were recruited using Prolific. All online workers in the United States who were 18 years of age and older were eligible to participate. Participants provided informed consent and were welcome to withdraw their participation in the study, for any reason at any point during the study, without penalty or loss of compensation. Data was collected between April 11 and May 25, 2022. Participants received $15 CAD in compensation.

A total of 212 dyads provided complete data. Four dyads were removed prior to analysis for not following instructions (i.e., did not use conversation-starter questions). The final dataset included 208 dyads or $N = 416$ participants (189 men; 216 women; 8 non-binary; 3 gender not listed; 2 prefer not to say). The mean age of our participants was 37.92 ($SD = 13.29$; range: 18-84).

**Study 2 Materials and measures**. Colloquially referred to as 'fast friends', this procedure was developed for experimentally manipulating closeness among strangers in a laboratory setting[3]. In the original experimental condition, stranger dyads alternated in asking each other a list of 36 increasingly personal questions over a 45-minute face-to-face conversation. Reciprocal disclosure of personal details led to increased interpersonal closeness, relative to a small-talk condition[3]. For the purposes of our study, we selected six increasingly personal questions for our stranger dyads to discuss. We referred to these questions as conversation-starters. Specifically, the following questions were used:

- What would constitute a "perfect" day for you?
- If you could change anything about the way you were raised, what would it be?
- Is there something that you've dreamed of doing for a long time? Why haven't you done it?
- What is your most treasured memory?
- If you were going to become a close friend with your partner, please share what would be important for him or her to know.
- Share with your partner an embarrassing moment in your life.

Following the conversation, participants completed several manipulation checks, and scales relating to their experience, evaluation of their partner, and their personality. Of particular interest was a measure of perceived honesty of their partner, which served as an index of 'deceiver's distrust' (i.e., our mediator), and two single-item scales measuring interconnectedness and closeness with their partner, which served as outcome measures of interpersonal closeness. We also included trait measures of deception frequency and loneliness. Due to experimenter error, these scales were added mid-way through data collection. Accordingly, data on these scales is not available for all participants. Each of these measures are described in detail, below.

The *Reysen Honesty Scale*[30] is an 8-item measure which provides a measure of the extent to which an individual is perceived as honest and includes items such as "I believe what this person says" and "I trust this person will tell me the truth." Items were rated on a 7-point Likert scale and showed high internal reliability ($\alpha = 0.91$).

The *Inclusion of Other in Self (IOS)*[31] measure is a single-item measure of interpersonal closeness which uses a series of images of overlapping circles representing the self and the conversational partner. A set of Venn-like diagrams, ranging from completely separate to completely overlapping circles make up this pictorial 7-point scale. This measure of interpersonal closeness was designed to tap into people's sense of being interconnected with another person. The scale has demonstrated strong convergent validity with lengthier measures of closeness (i.e., *Relationship Closeness Inventory*)[32] and has been used to validate the experimental generation of closeness, using the guided

conversation questions that we employed in this study[3]. As a supplementary measure of closeness, we also included one face-valid item, "On the following scale, please rate how close you feel to your partner", which participants answered on a 7-point Likert scale from 1 (not at all) to 7 (very).

Additionally, participants were asked to rate the extent to which they engaged in deception during the discussion of each topic-question that guided their conversation. Specifically, they were asked: "Please rate the extent to which your answers to each of the following questions involved deception (i.e., lies meant to mislead your partner)" followed by a list of the topic-questions, and a 1(not at all deceptive) to 5 (completely deceptive) scale with an option to indicate that they did not discuss this topic question ("Ran out of time before we got to this question"). A mean of responses on these items (excluding "Ran out of time …" responses) was calculated to provide a measure of deception by each participant.

The *Lying in Everyday Situations (LiES)*[33] survey consists of 14-items and provides a measure of the extent to which people participate in deception in their daily lives (overall score: $\alpha = 0.90$). Two 7-item subscales tap the use of vindictive lies ($\alpha = 0.93$) and relational lies ($\alpha = 0.91$), specifically. Vindictive lies are generally told to harm others or benefit the self (e.g., "I lie for revenge") while relational lies were told to maintain social cohesion (e.g., "I tell lies in order to spare another's feelings"). These scales showed high test-retest reliability and strong convergent validity with related scales, including Machiavellian personality traits and self-reports of deception frequency[33].

The *UCLA Loneliness-8* (ULS-8)[34] scale is a short-form version of the 20-item UCLA Loneliness Scale (ULS-20)[35]. This 8-item scale provides a brief measure of the subjective sense of loneliness —a deficiency in social contact, relative to what is desired. This short-form version of the scale showed high internal reliability ($\alpha = 0.91$) in our sample and previous research suggests that it is highly correlated with the original 20-item version ($r = 0.91$)[34], while reducing participant burden.

Participants were also asked to complete the *Relational Communication Scale*[36], rate their partner on basic dimensions of social evaluation (e.g., warmth, competence, morality)[37], likeability, and indicate whether they thought their partner was lying to them. It is worth noting that only $n = 10$ of 209 (i.e., 4.8%) of the receivers responded 'yes' to the question, "At any point during the conversation did you think that your partner was lying to you?", indicating a strong truth bias among receivers[24]. A full list of measures and the order in which they were asked can be found in the Qualtrics file, posted to OSF (https://osf.io/ezn7p/). These measures were included to test a separate series of hypotheses and are not examined further here.

**Study 2 Procedures**. Participants completed the consent form and were provided instructions about their upcoming conversation. All participants were told they would be paired with a random stranger for a chat-based conversation using ChatPlat software (www.chatplat.com) embedded in the Qualtrics survey. ChatPlat provided the capacity to match senders with receivers in real time, a forum for text-based conversations to occur, and recording of text data. Participants were told that they would be provided with six conversation-starter questions to guide the discussion. Participants were asked to imagine that they were considering their conversation partner as a potential roommate and to use this conversation to get to know them to determine whether they would be a good fit. They were encouraged to ask follow-up questions and to move on to the next conversation-starter when the previous topic had been exhausted. Participants were also asked to keep the nature of their conversation confidential.

Participants completed several attention-check questions to ensure that they understood these instructions; they could not move forward until they had answered all questions correctly. Participants were then provided with a list of the conversation-starter questions and were given two minutes to review them before proceeding.

Participants were then randomly assigned to be Participant 1 who would always answer the conversation-starter questions first, or Participant 2 who would answer second. For ease of description, we refer to Participant 1 as the sender, and Participant 2 as the receiver, throughout the manuscript. Senders were further randomly assigned to be as complete, open, and honest as possible, or to lie to their partner for the entirety of the conversation. Senders in the lie condition were assured that these were secret instructions and were asked to be as convincing as possible. All receivers were asked to be as complete, open, and honest as possible.

Participants then entered the chat where a sender and receiver were always paired together. Dyads were instructed to chat until they discussed all six questions, or up to 25 minutes (whichever came first). Participants' conversations ranged from 5.16 minutes to 28.25 minutes, with an average duration of 21.74 minutes ($SD = 4.97$). An independent samples $t$-test yielded no statistically significant evidence that senders responded to a different number of questions between the truth ($M = 4.38$, $SD = 1.69$) and lie ($M = 4.49$, $SD = 1.40$) conditions, $t(206) = 0.51$, $p = 0.610$, $d = 0.07$ [95% CI: −0.34, 0.20]. We used JABAApprox.1 function in R to approximate the Bayes Factor, finding a BF of 18.02, indicating strong support for the null hypothesis. After exiting the chat, participants were asked if they were successfully paired with another participant and able to complete the chat session. If not, they were able to exit the study while still receiving compensation for their time. For those who were successfully paired, they proceeded to complete measures about their partner, their experience in the conversation, and themselves. Participants also provided demographic information, were debriefed about the true nature of the study, and were provided compensation. This study was approved by the University of British Columbia, Okanagan Behavioural Research Ethics Board.

In our final study, we further considered whether the dispositional tendency to tell lies is associated with a general distrust of others and a sense of loneliness. In Study 3, we sought to gather a larger sample than in Study 2, include additional control variables, and examine the mediating role of dispositional interpersonal trust. Thus far, loneliness has been characterized by an individual's subjective lack of social connection, however, in Study 3 we examine the role of lacking social connection objectively. Studies including objective network characteristics such as the number of friends, relatives, and frequency of contact with social network members show an inverse relationship with loneliness[38–41]. Specifically, in Study 3 we included objective social network characteristics (i.e., social network size and diversity) as control variables in our analyses. As before, we expected that lies would be associated with a sense of loneliness, even when accounting for the number and type of close social ties [H1]. Additionally, we examined whether interpersonal trust mediated the relationship between dispositional use of deception and the experience of loneliness [H2].

**Study 3 Participants**. We recruited participants using Prolific. All online workers in the United States and 18 years of age or older were eligible to participate (participant age data is unavailable due to experimenter error). This study received approval from the Ethics board at the University of British Columbia. Participants provided informed consent and could withdraw participation in

the study for any reason, at any point during the study, without penalty or loss of compensation. A power analysis indicated that $N = 395$ participants would be necessary to find a small effect ($f^2 = 0.02$) in a multiple regression with three predictors, setting $p = 0.05$ and $1-\beta = 80\%$. $N = 399$ participants completed the study, and no data exclusions were made. Of these individuals, 178 participants identified as men, 206 as women, 11 as non-binary, and 4 selected 'prefer not to say'. One participant selected both 'woman' and 'non-binary'. Data was collected on October 13, 2022. Prolific participants received $3.00 CAD for their time.

**Study 3 Measures**. Participants completed the following four scales: Lying in Everyday Situations Scale (LiES)[3], General Trust Scale (GTS)[42], the revised UCLA Loneliness Scale (ULS-20)[35], and the Social Network Index Revised (SNI)[43].

The *UCLA Loneliness* Scale (ULS-20)[35] includes 20 items (e.g., No one really knows me well), each rated on a 1 (never) to 4 (often) scale. It is widely used to measure loneliness and a mean score of all items has demonstrated strong convergent and discriminant validity with related constructs[35]. This measure showed high internal consistency in our sample ($\alpha = 0.92$).

The *General Trust Scale* (GTS)[42] is a 6-item measure and uses general statements to measure beliefs about the honesty and trustworthiness of others. Each item is rated on a 1 (strongly disagree) to 7 (strongly agree) scale. This measure showed high internal consistency in our sample ($\alpha = 0.87$).

The *Social Network Index Revised* (SNI)[43] measure asks participants to report on participation in 12 types of social relationships (e.g., spouse, parents, parents-in-law, children, other close family members, close neighbors, friends). To assess network diversity, participants are assigned one point for each relationship type (maximum score: 12) for which they indicate they speak to a person (or multiple people) fitting that description at least once every two weeks. We also calculated social network size by summing across the 12 roles. To ensure that this figure was not artificially inflated by individuals reporting large group memberships (e.g., of volunteer organizations), we followed the recommendation of recoding very large groups to have an upper bound of seven[43].

**Study 3 Procedure**. Participants provided informed consent and completed the Lying in Everyday Situations Scale (LiES)[3], General Trust Scale (GTS)[42], the revised UCLA Loneliness Scale (ULS-20)[35], and the Social Network Index Revised (SNI)[43] in random order. Participants also provided basic demographic information before being debriefed, thanked for their time, and provided compensation. This study was approved by the University of British Columbia, Okanagan Behavioural Research Ethics Board.

**Reporting summary**. Further information on research design is available in the Nature Portfolio Reporting Summary linked to this article.

## Results

**Study 1**. Using the lmerTest[44] package in R, a mixed effects regression model was conducted to account for interdependence in dyads. Specifically, ratings of trustworthiness were included as a fixed effect predictor, dyads as a random effect predictor, and feelings of closeness as the outcome. Data distribution was assumed to be normal but this was not formally tested. Consistent with H1, participants' who rated themselves as relatively untrustworthy also reported reduced feelings of closeness with their partner, b = 0.25, 95% CI [0.15, 0.35], $SE = 0.05$, $t(758.61) = 4.85$, $p < 0.001$.

To test whether perceived partner trustworthiness mediated this relationship, we used MLmed[45] in SPSS to conduct a multi-

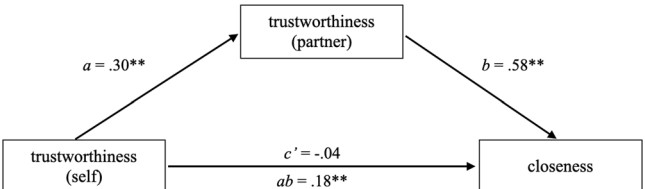

**Fig. 1 A multi-level mediation model testing the effect of self-rated trustworthiness on perceived closeness with conversational partner through perceived partner trustworthiness.** *$p < 0.05$, **$p < 0.01$. We used MLmed[45] in SPSS to conduct a multi-level mediation model, nesting Level-1 variables within dyads ($n = 776$ individuals in 388 dyads) The indirect effect was statistically significant ($ab = 0.18$, 95% CI: 0.11, 0.25).

level mediation model, nesting Level-1 variables within dyads. As predicted in H2, the indirect effect was statistically significant ($ab = 0.18$, 95% CI: 0.11, 0.25). Coefficients presented in Fig. 1 indicate that participants' ratings of their own trustworthiness were positively related to perceptions of their partner's trustworthiness, which was associated with increased feelings of closeness. A second mediation model was also considered, in which closeness ratings were proposed to mediate the relationship between self-ratings of trustworthiness and perceived partner trustworthiness. This analysis produced a marginally significant indirect effect ($ab = 0.06$, 95% CI: 0.00, 0.12). Findings are consistent with the hypothesis that participants who rated themselves as relatively untrustworthy saw their partner similarly, and also reported feeling less close to them.

**Study 2**. To test whether participants assigned to sender role complied with the veracity manipulation, an independent samples $t$-test was conducted comparing responses to manipulation check questions across the truth and lie conditions. As instructed, participants assigned the 'sender' role reported being more deceptive in the lie ($M = 4.51$; $SD = 0.66$) versus the truth condition ($M = 1.25$; $SD = 0.75$), $t(206) = 33.34$, $p < 0.001$, $d = 4.63$, 95% CI [4.10, 5.15]. Consistent with the instructions that they received, receivers reported telling very few lies in their conversation and there was no statistically significant evidence that receivers differed in their honesty, regardless of whether they were paired with a truthful ($M = 1.20$; $SD = 0.59$) or deceptive sender ($M = 1.13$; $SD = 0.36$), $t(205) = 0.97$, $p = 0.33$, $d = 0.14$ [95% CI: $-0.14$, 0.41]. We used JABApprox.1 function in R to approximate the Bayes Factor, finding a BF of 12.99, indicating strong support for the null hypothesis.

Using the lmerTest[44] package in R, a pair of mixed effects regression models were conducted to account for interdependence in dyads. Specifically, veracity (truth vs. lie), role type (sender vs. receiver) and a veracity x role type interaction term were included as fixed effects, and dyads as a random effect, with IOS and closeness ratings as the outcomes. Veracity and role type were effect coded ($-0.5$, 0.5) to aid interpretation of coefficients. Data distribution was assumed to be normal but this was not formally tested. Table 1 provides all coefficients. For both measures, there was a significant effect of veracity, such that participants in the lie condition felt less connected (IOS) and less closeness with their partner [H1; see Fig. 2]. Interestingly, we found no statistical evidence of an interaction effect.

Because we were primarily interested in the effect of telling lies on senders' experience of interconnectedness and closeness, we focused our mediation analyses on these participants. Specifically, we used PROCESS[46] (5000 bootstrap samples) in SPSS to test whether distrust, as measured using the Reysen Honesty Scale[30], mediated the relationship between sender veracity and interconnectedness (IOS). Consistent with H2, the indirect effect was

**Table 1 Coefficients of linear mixed effect models, testing the effects of veracity and role on ratings of interconnectedness and closeness within conversational dyads.**

| Fixed effects | Model 1: IOS | Model 2: closeness |
|---|---|---|
| Intercept | 3.648** | 4.236** |
| Veracity | −0.325* | −0.362* |
| (−0.5 = truth; 0.5 = lie) | | |
| Role Type | −0.020 | 0.189 |
| (−0.5 = sender; 0.5 = receiver) | | |
| Veracity x Role Type | 0.021 | 0.082 |

*$p < 0.05$, **$p < 0.001$.

significant, $ab = -0.41$, 95% CI [−0.63, −0.21]. Coefficients presented in Fig. 3 illustrate that deceptive senders rated their partner as less honest and perceived honesty of the partner was positively related to interconnectedness ratings. Similarly, a mediation model that replaced interconnectedness with closeness as the outcome revealed a similar, significant, indirect effect, $ab = -0.48$, 95% CI [−0.74, −0.25] (see Fig. 3 for additional details). Two additional mediation models were conducted, which reversed the order of the mediator (perceived honesty of receiver) and outcomes (IOS, closeness). Here, we found that closeness did not mediate the relationship between sender veracity and the perceived honesty of the receiver ($ab = -0.14$ [−0.28, 0.00]). Similarly, IOS did not mediate the relationship between sender veracity and the perceived honesty of the receiver ($ab = -0.10$ [−0.21, 0.02]).

A set of Pearson bivariate correlations were conducted to examine the association between responses on the *Lying in Everyday Situations* (LiES) scale (overall score, vindictive and relational sub-scales) and the *UCLA Loneliness-8* (ULS-8) measure of loneliness. Consistent with H1, greater overall LiES scores were positively associated with self-reported loneliness, $r(278) = 0.27$, $p < 0.001$, 95% CI [0.157, 0.375]. Use of relational lies, in particular, was positively associated with self-reported loneliness, $r(280) = 0.28$, $p < 0.001$, 95% CI [0.168, 0.385], while the relationship between vindictive lies and loneliness did not reach statistical significance in this sample, $r(280) = 0.10$, $p = 0.086$, 95% CI [−0.015, 0.217]. For the latter relationship, we also used JABAApprox.1 function in R to approximate the Bayes Factor, finding a BF of 5.51, indicating strong support for the null hypothesis.

**Study 3.** A series of Pearson correlations between dispositional use of deception, trust, loneliness, and social network characteristics were conducted (see Table 2). Data distribution was assumed to be normal but this was not formally tested. Figure 4 provides an illustration of the positive relationship between telling lies (overall LiES score) and self-reported loneliness (ULS-20 Loneliness). Consistent with Study 2 and H1, the overall LiES score and vindictive lies subscale were positively associated with loneliness. In this larger sample, the relational lies subscale was also positively associated with loneliness.

To test whether the relationships between dispositional use of deception and loneliness persisted when accounting for individuals' social network characteristics, three multiple regression analyses were conducted. First, overall LiES score, number of close contacts, and diversity of close contacts were regressed on self-reported loneliness. The model was significant, $F(3, 395) = 247.51$, $p < 0.001$, explaining 27% ($R^2 = 0.27$) of the variance in loneliness (ULS-20[35]) scores. As expected, the

dispositional use of deception was positively associated with loneliness, $\beta = 0.28$, $p < 0.001$, 95% CI [0.13, 0.24], over and above social network characteristics.

We conducted the same analysis substituting the subscales for the overall LiES score to assess if the positive relationship held for both vindictive and relational lies. With respect to vindictive lies, the model was significant, $F(3, 395) = 38.16$, $p < 0.00$, explaining 23% ($R^2 = 0.23$) of the variance in the outcome variable. Vindictive lies accounted for significant variance in self-reported loneliness, $\beta = 0.20$, $p < 0.001$, 95% CI [0.09, 0.22], with social network characteristics included in the model. Finally, the model testing the role of relational lies in self-reported loneliness was also significant, $F(3, 395) = 44.55$, $p < 0.001$, explaining 25% ($R^2 = 0.25$) of the variance. Relational lies accounted for significant variance in loneliness, $\beta = 0.26$, $p < 0.001$, 95% CI [0.07, 0.14], over and above social network characteristics.

We conducted a series of three PROCESS (5000 bootstrap samples) models to test whether participants' predisposition to trust mediated the relationship between the dispositional use of deception (overall, vindictive, relational) and loneliness. As predicted in H2, the indirect effect was significant for all models: overall ($ab = 0.04$, 95% CI [0.02, 0.07]; see Fig. 5), vindictive ($ab = 0.04$, 95% CI [0.01, 0.07]), and relational lies ($ab = 0.02$, 95% CI [0.01, 0.04]). In all models, the use of deception (overall, vindictive, or relational) was negatively associated with trust, which was negatively associated with self-reported loneliness. We also conducted mediation models that reversed the order of our proposed mediator and outcome. Here, we found that loneliness mediated the relationship between the disposition to lie and interpersonal trust. This was true for lies overall ($ab = -0.09$ [−0.14, −0.06]), vindictive ($ab = -0.09$ [−0.13, −0.05]), and relational lies ($ab = -0.05$ [−0.08, −0.03]).

## Discussion

Discovering you have been deceived can breed distrust and damage relationships[47,48]. The current research suggests that detection is unnecessary for this outcome to occur; from the perspective of the lie-teller, distrust occurs immediately and social connection may never form, even when interacting with an honest person. Across three studies, we find that relatively untrustworthy interlocutors, people randomly assigned to lie in a conversation, and people who report using deception more than most in their everyday lives all report decreased social connection (i.e., less closeness, increased loneliness), relative to honest actors. We also find that these effects appear to be mediated by 'deceivers' distrust'.

**Study 1.** Our first study used an existing database of free-form conversations between strangers[28] to find that people who rated themselves as relatively less trustworthy also reported feeling less close to their conversational partner after a 25-minute video-based discussion. Additionally, analyses revealed that this relationship may be mediated by perceived partner trustworthiness; people who rated themselves as less trustworthy perceived their partners as relatively untrustworthy too, which was associated with a decreased sense of closeness between partners.

Using a pre-existing conversational database (CANDOR corpus)[28], Study 1's findings demonstrate initial support for the notion that relatively untrustworthy people are less likely to forge strong social connections, and that deceiver's distrust may provide an explanation for this relationship. That said, the correlational nature of this study precludes our ability to establish causality and follow-up analyses suggest that the role of our proposed mediator and outcome may be reversed. Specifically,

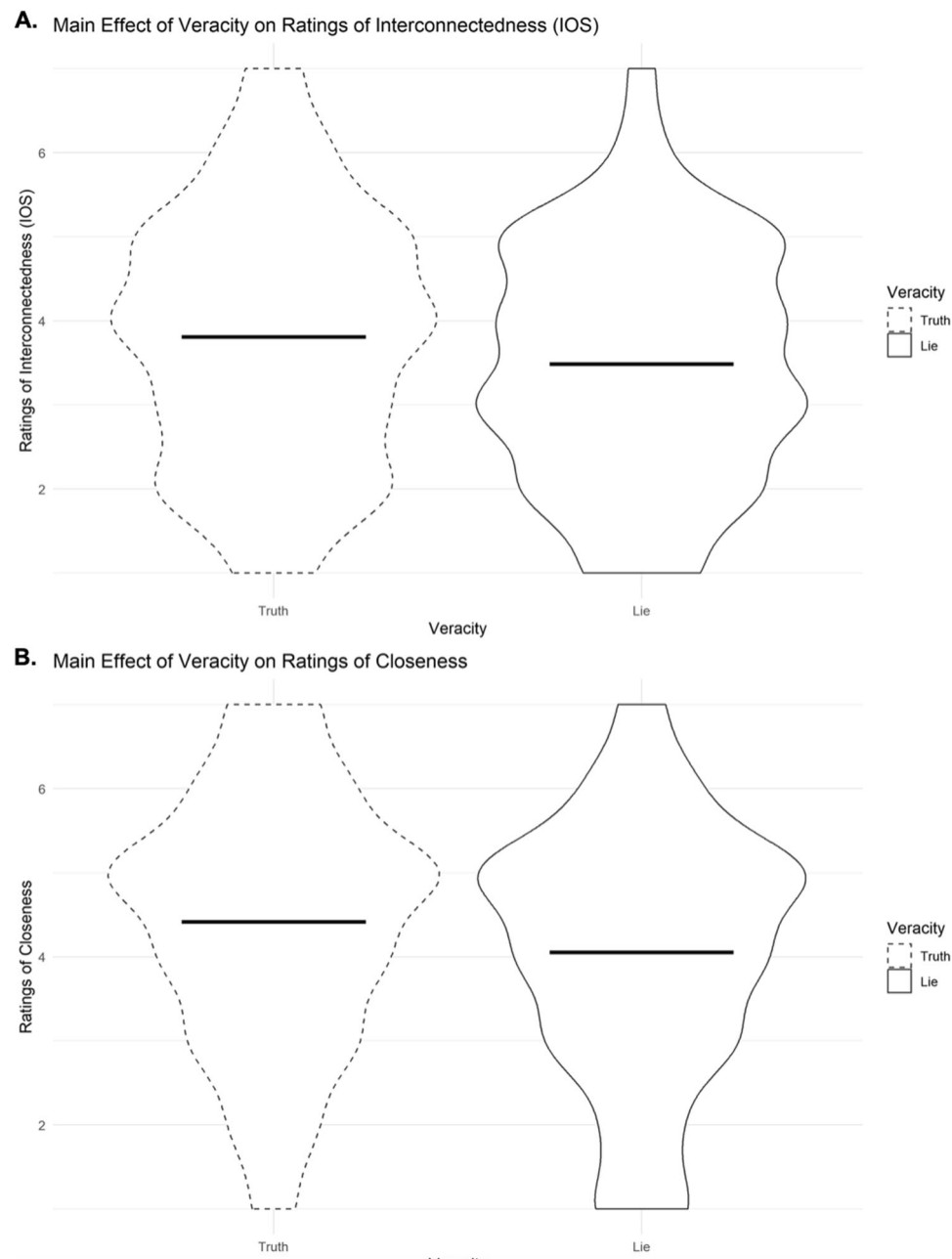

**Fig. 2 Violin plots illustrating the main effect of veracity on measures of interpersonal closeness.** Means are indicated by horizontal lines; curved lines indicate data distribution. Panel **A** displays the main effect of veracity on perceived interconnectedness (b = −0.325, p = 0.028, 95% CI [−0.613, −0.037]), and panel **B** reflects the main effect of veracity on closeness (b = −0.362, p = 0.022, 95% CI [−0.668, −0.056]) among n = 416 individuals in 208 dyads.

closeness ratings also mediate the relationship between self-reported trustworthiness and perceived partner trustworthiness.

**Study 2**. By experimentally manipulating veracity, Study 2 built on this dyadic conversational paradigm to provide causal evidence that deception reduces closeness. Participants who were randomly assigned to lie (vs. tell the truth) during a 25-minute text-based conversation with a stranger reported feeling less closeness with their partner. This suggests that lying can cause people to feel less close to their conversational partners, relative to telling the truth. A mediation analysis provided support for perceived trustworthiness as a mediator of veracity and feelings of closeness. This analysis suggested that deceptive senders rated their conversational partners as less honest than truthful senders, and that this was associated with decreased closeness.

Interestingly, we found no statistically significant evidence that the role to which participants were assigned (sender or receiver) affected ratings of closeness, nor did we find any evidence of an interaction between role assignment and veracity condition. This suggests that receivers may also experience a relative decrease in closeness when partnered with a liar (vs. truth-teller). Although the receiver's experience was not a focus of this investigation, these findings call for future research on how deception influences the course of a conversation and how lies are perceived by naïve receivers, who are rarely suspicious about the possibility of deception.

Closeness and interconnectedness did not mediate the relationship between sender veracity and perceived honesty of the receiver, suggesting that—at least in this experimental context—our proposed mediation model is a better explanation of the data than an alternative in which the mediator and outcome variables are reversed. We also found initial evidence that the frequency with which people tell lies is associated with their self-reported loneliness. Specifically, participants who report telling more lies overall, and more vindictive lies, also reported experiencing greater loneliness.

**Study 3**. Finally, we provide evidence that a connection between deception and loneliness can be observed in dispositional measures and that this relationship may be mediated by participants' disposition to trust. Individuals who report using deception for both vindictive and relational reasons also report experiencing greater loneliness in their lives, even when controlling for their social network size and diversity. It is particularly interesting that relational lies show this pattern, given that they are told with the express purpose of protecting social relationships. That is, even when lies are told to escape conflicts or spare others' feelings, they are associated with feelings of loneliness. These findings build on previous research which found that people overestimated the benefits of kindness and underestimated the costs of honesty with respect to social connection[18]. In addition, findings suggest that individuals who report increased dispositional use of deception

(overall, vindictive, or relational) are less likely to trust others, mediating the relationship with loneliness. Our data were also consistent with the hypothesis that interpersonal trust would mediate the relationship between lying and loneliness. However, our data also supported alternative mediation models—specifically, that loneliness mediates the relationship between lying and interpersonal trust.

**Limitations**. Future research should seek to further understand the robustness of this effect by altering the laboratory methodology. The veracity manipulation employed in Study 2 was rather blunt, as individuals were instructed to tell the truth or lie for the entirety of the conversation. Future research should explore more ecologically valid manipulations or situations—perhaps having participants answer dishonestly to only a select number of questions or leading some participants to lie without explicitly instructing them to do so. In order to provide additional ecological validity, future work may consider using a face-to-face context. Since there is some evidence that deception may be expressed and evaluated differently in close relationships (vs. stranger dyads)[49], future research should also examine how lies impact perceived closeness in already established relationships. Generalizability may also be improved by studying individuals from other countries and cultures. Our data are limited to residents of the United States—a western, industrialized, and individualistic nation.

Future work should also consider additional psychological mechanisms underpinning the relationship between deception and social connection. While we provide evidence that 'deceiver's distrust' appears to mediate the relationship between deception and closeness in an experimental context, the data in correlational Studies 1 and 3 are also consistent with the possibility that lying decreases trust by reducing social connection. In other words—the relationship between deceivers' distrust and feelings of closeness or loneliness may be bidirectional. Other mechanisms too may be worth testing. For example, the cognitively taxing experience of telling a lie may be subjectively experienced as disfluency and misattributed to evaluations of the conversational partner[50]. Although future research may provide additional insight into mechanisms, current findings are consistent with meta-analytic findings that interventions targeting maladaptive social cognition can reduce loneliness and suggest that focusing on biases that affect the perceived trustworthiness of conversational partners, including one's own trustworthiness, may improve people's ability to forge social connections[51].

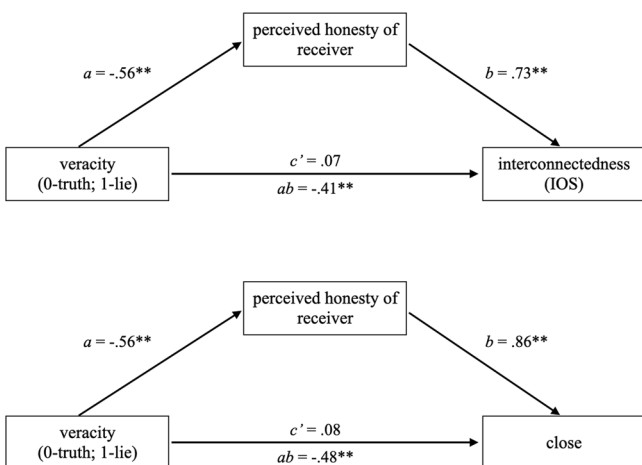

**Fig. 3 Mediation models testing the effect of sender veracity on perceived interconnectedness (IOS) and closeness with receiver through perceived receiver honesty.** *$p < 0.05$, **$p < 0.01$. We used PROCESS[46] (5000 bootstrap samples) in SPSS to test whether distrust mediated the relationship between sender veracity and interconnectedness ($n = 416$ individuals in 208 dyads). The indirect effect was significant, $ab = -0.41$, 95% CI [−0.63, −0.21].

## Conclusion
Although there is much yet to learn about the relationship between deception and social connection, our findings may have practical application in relationship counseling, mental health therapy, and initiatives to improve well-being. Trust has been on

**Table 2 Pearson correlations between dispositional use of deception, trust, loneliness, and social network characteristics.**

| | 1. | 2. | 3. | 4. | 5. | 6. | 7. |
|---|---|---|---|---|---|---|---|
| 1. LiES - Relational | (0.90) | | | | | | |
| 2. LiES - Vindictive | 0.366** | (0.88) | | | | | |
| 3. LiES - Overall | 0.906** | 0.725** | (0.89) | | | | |
| 4. SNI – Diversity of Contacts | −0.088 | −0.144** | −0.131** | (−) | | | |
| 5. SNI – Number of Close Contacts | −0.097 | −0.120* | −0.126* | 0.783** | (−) | | |
| 6. GTS – Trust | −0.172** | −0.162** | −0.200** | 0.234** | 0.174** | (0.87) | |
| 7. ULS-20 – Loneliness | 0.298** | 0.258** | 0.337** | −0.426** | −0.373** | −0.368** | (0.92) |

Cronbach alpha reliabilities appear on diagonal.
*$p < 0.05$, **$p < 0.001$.

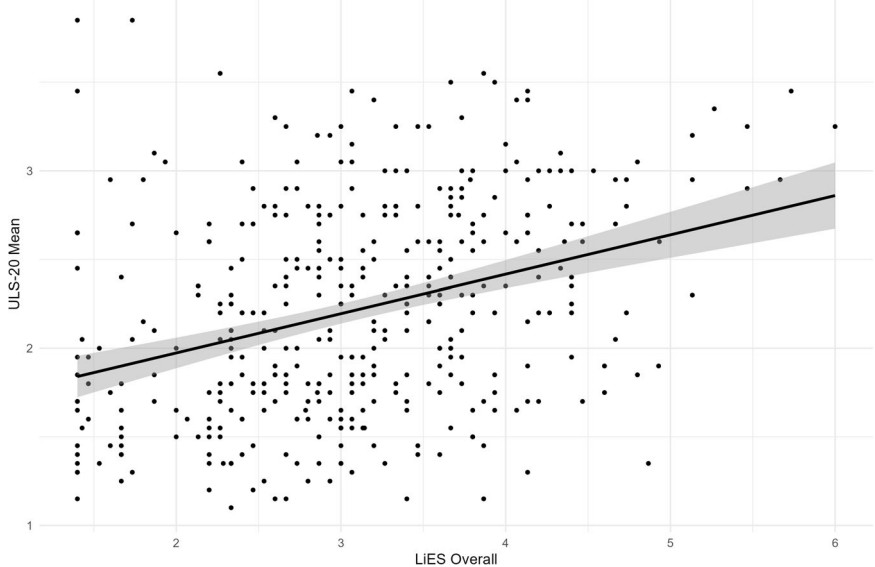

**Fig. 4 A scatterplot illustrating the relationship between lying behavior (LiES overall) and self-reported loneliness (ULS-20 loneliness).** Linear trendline is provided; grey area indicates 95% confidence region. A Pearson correlation ($n = 399$ individuals) revealed a significant positive association between participants' average self-reported loneliness and lying behaviour ($r$ (397) = 0.337, $p < 0.001$, CI [0.25, 0.42]). The x-axis displays lying behaviour, measured by the Lying in Everyday Situations Scale[33], while average self-reported loneliness, measured by the 20-item UCLA Loneliness Scale[34], is on the y-axis.

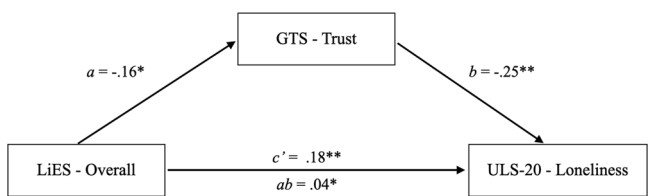

**Fig. 5 A mediation model testing the relationship between lying behavior (LiES overall) and self-reported loneliness through interpersonal trust.**
*$p < 0.05$, **$p < 0.01$. We used PROCESS[46] (5000 bootstrap samples) model in SPSS to test whether participants' predisposition to trust mediated the relationship between the dispositional use of deception and loneliness ($n = 399$ individuals). The indirect effect was significant ($ab = 0.04$, 95% CI [0.02, 0.07]). Deceptive behaviour is measured using the Lying in Everyday Situations Scale (LiES)[33], while self-reported loneliness is measured by the 20- item UCLA Loneliness Scale (ULS-20)[34]. Interpersonal trust is measured by the General Trust Scale (GTS)[42].

the decline in the U.S. and around the world for decades, and the U.S. Surgeon General recently identified loneliness as a public health crisis[52,53]. Our findings suggest that learning to engage in honest conversations—even when they may be difficult or uncomfortable—may provide an avenue for improving social relationships and well-being, more generally.

While previous research has highlighted the power of conversation to generate social connection, we highlight an important moderator of this effect: honesty. Dishonesty, it seems, is detrimental to the sender's well-being—breeding distrust and diminishing social connection. Findings underscore the consequences of deception in social life, even when undetected, and provide support for the old adage that honesty is the best policy.

## Data availability
Data used in Study 1 analysis is available for download from BetterUp Inc. here: https://betterup-data-requests.herokuapp.com/. The Qualtrics surveys and deidentified data for Studies 2 and 3 are also available on OSF: https://osf.io/ezn7p/.

## Code availability
Data analysis code is available on OSF: https://osf.io/ezn7p/.

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

## Acknowledgements

This research was funded by an Insight Grant (435-2022-0389) from the Social Science and Humanities Research Council of Canada, awarded to the last author. The funders had no role in study design, data collection and analysis, decision to publish or preparation of the manuscript.

## Author contributions

S.S. was involved in conceptualization of study design, methodology, data collection, data analysis, writing, and manuscript preparation. C.B. was involved in data analysis, writing, manuscript preparation, and dissemination efforts. L.tB. supervised both S.S. and C.B., and led project administration, conceptualization of study design, methodology, data collection, data analysis, writing, and manuscript preparation.

## Competing interests

The authors declare no competing interests.
