## [Peer Review File · Communications Psychology]

4th May 23

Dear Dr ten Brinke,

Thank you for your patience during the peer-review process. Your manuscript titled "Feeling Alone: A Novel, Social Consequence of Telling Lies" has now been seen by 3 reviewers, and I include their comments at the end of this message. They find your work of interest, but raised some important points. We are interested in the possibility of publishing your study in Communications Psychology, but would like to consider your responses to these concerns and assess a revised manuscript before we make a final decision on publication.

We therefore invite you to revise and resubmit your manuscript, along with a point-by-point response to the reviewers. Please highlight all changes in the manuscript text file.

Editorially, we consider it important to address the reviewers' concerns regarding the interpretation of the cross-sectional, correlational findings, which do not allow for directionality or casualty claims. Please remove all causal language from the description and interpretation of correlational data, analyses, and results.

Further, we ask that you address the methodological concerns, especially those raised by Reviewer #2, through additional analysis of the data. Please do not remove any study from the main manuscript. Any remaining ambiguities which cannot be ultimately resolved through additional analysis must be transparently discussed as a limitation. Finally, we ask that you revise your literature review to incorporate important missing publications.

Please use the following link to submit your revised manuscript, point-by-point response to the referees' comments (which should be in a separate document to any cover letter) and the completed checklist:
[link redacted]

Please do not hesitate to contact me if you have any questions or would like to discuss these

revisions further. We look forward to seeing the revised manuscript and thank you for the opportunity to review your work.

Best regards,

Jennifer Bellingtier

Jennifer Bellingtier, PhD
Senior Editor
Communications Psychology

EDITORIAL POLICIES AND FORMATTING

Editorial Policy: [Policy requirements](https://www.nature.com/documents/nr-editorial-policy-checklist.pdf) (Download the link to your computer as a PDF.)

Furthermore, please align your manuscript with our format requirements, which are summarized on the following checklist:

[Communications Psychology formatting checklist](https://www.nature.com/documents/commsj-style-formatting-checklist-review-perspective.pdf)

and also in our style and formatting guide [Communications Psychology formatting guide](https://www.nature.com/documents/commspsychol-style-formatting-guide-accept.pdf) .

* **CODE AVAILABILITY:** All Communications Psychology manuscripts must include a section titled "Code Availability" at the end of the methods section. In the event of publication, we require that the custom analysis code supporting your conclusions is made available in a publicly accessible repository; at publication, we ask you to choose a repository that provides a DOI for the code; the link to the repository and the DOI will need to be included in the Code Availability statement. Publication as Supplementary Information will not suffice. We ask you to prepare code at this stage, to avoid delays later on in the process.

* **DATA AVAILABILITY:**

All Communications Psychology manuscripts must include a section titled "Data Availability" at the end of the Methods section or main text (if no Methods). More information on this policy, is available at <http://www.nature.com/authors/policies/data/data-availability-statements-data-citations.pdf>.

At a minimum the Data availability statement must explain how the data can be obtained and whether there are any restrictions on data sharing. Communications Psychology strongly endorses open sharing of data. If you do make your data openly available, please include in the statement:

We recommend submitting the data to discipline-specific, community-recognized repositories, where possible and a list of recommended repositories is provided at <http://www.nature.com/sdata/policies/repositories>.

If a community resource is unavailable, data can be submitted to generalist repositories such as [figshare](https://figshare.com/) or [Dryad Digital Repository](http://datadryad.org/). Please provide a unique identifier for the data (for example a DOI or a permanent URL) in the data availability statement, if possible. If the repository does not provide identifiers, we encourage authors to supply the search terms that will return the data. For data that have been obtained from publicly available sources, please provide a URL and the specific data product name in the data availability statement. Data with a DOI should be further cited in the methods reference section.

REVIEWERS' EXPERTISE:

Reviewer #1 Deception, Interpersonal communication

Reviewer #2 Deception, Morality

Reviewer #3 Deception, Interpersonal relationships

REVIEWERS' COMMENTS:

Reviewer #1 (Remarks to the Author):

This paper tells a good story. The idea that one's own lying leads to own loneliness independent of detection is a nifty idea that is supported by three studies. Of course, the effect should be much stronger for detected lies about important things which is a good way to get oneself ostracized.

In the section "A False Consensus of Distrust" (In 87-102), the current text might create the mistaken

impression that the idea is original to the current paper. Markowitz and Hancock (2018; doi.org/10.1093/joc/jqy019) call this the “False Consensus Effect for Deception.” They should be cited and credited.

In figures 1 and 2, the arrows from trustworthiness to closeness and perceived honesty to IOS/close (i.e., mediator to outcome) could go either direction. The closer I am to someone, the more I tend to trust them and believe them. That is, both own honesty and closeness to other could predict trust and perceived honesty and fit the results equally well. This, I think, makes the mediation argument weak.

There might be an alternative explanation for the honesty effects in study two where veracity was manipulated. Participants in the lie condition might presume that their partner also might be told to lie. At the very least, the experimental instructions in the lie condition might act as a trigger from the perspective of truth-default theory.

Minor things:

Ln 45-46 “For example, people also admit to telling an average of two lies per day, with some individuals engaging in many more,” while this is literally true, it is misleading because the distribution is highly skewed. Yes some people are above average, but most people are below the average. The 2 lies were for students; non-students averaged one with a median of less than one lie every other day. Also see, doi.org/10.1111/j.1468-2958.2009.01366.x.

For convergent validity of the IOS, also see: doi.org/10.1037/a0026265

In the McCornack & Levine (1990) study, my memory is that negative outcomes were more what the lie was about than the mere act of lying. Lying about a surprise birthday party is different than lying in order to defraud someone.

Reviewer #2 (Remarks to the Author):

The authors examined dishonesty in social interactions and perceived trust, closeness and loneliness across three studies. The manuscript is easy to read and makes a novel contribution to the field. I have provided some thought form strengthening the manuscript below.

Abstract: The first two sentences of the abstract seem disconnected. It seems like two separate ideas and I think the authors could more clearly integrate the ideas and purpose of the study here.

The title focusses on Loneliness but wasn't examined in Study 1 and almost feels like a secondary component in Study 2 (as it wasn't part of the main experimental manipulation). The set of studies seem to largely focus on trust and closeness primarily. The loneliness findings are interesting and important but perhaps not as central as the title suggests. Furthermore, the title suggests that feeling alone is the consequence of telling lies, however, this temporal order was not assessed. Perhaps loneliness results in greater lie-telling. Given that only correlations of loneliness were examined the authors should be cautious with their language. This is also true for the mediation

analyses. Since all measures, in all studies, were taken at a single time point, the order could be revised in the model (particularly for study 3) and the same relation found. As such the authors should be cautious with causal or directional language throughout the manuscript.

Social Consequences of honesty: the authors note that others have not examined the social consequences of honesty, however, Dykstra has two studies examining the longitudinal associations between lying and secret keeping to friends and to parents and the associations with relationship quality and depressive symptoms. I think these papers (both in 2020) would help strengthen the authors' argument that lying has social consequences for the lie-teller. Furthermore, Dykstra argues that the bidirectional association between lying/secret keeping and depression is likely due to the lie-teller concealing depression but also lying isolating the lie-teller from their support system. These findings support the authors' ideas that lying is related to loneliness and that there are consequences for the lie-teller. While Dykstra examines adolescence, these findings appear to be relevant here. I believe DePaulo also has work showing that adults rate social interactions more negatively when they lie which would be relevant here.

A False Consensus of Distrust: the authors argue that one's own dishonesty influences their perception of others and cite research around the sender's perception of the deceiver. Another relevant citation here is Evans & Lee (2014) in which they examined adolescent truth and lie-tellers lie-detection of other's truths and lies and found that participants' own prior dishonesty biased their judgements. Again, I think this strengthens the authors' arguments here.

The authors reported that participants were to talk for 25 minutes – can they report how long participants ended up talking for? Mean length and range of time the conversations lasted? The authors sub heading is “Less Trustworthy Participants Felt Less Close to Conversational Partners” but the results are written such that more trustworthiness was positively related to closeness. Aligning these two would be helpful for the reader.

Before presenting mediation analyses the authors should confirm that all variables were correlated. Also, there are limitations to these conclusions since the authors collected all data at a single time point.

It would be helpful at the end of each study if the authors provided a mini-discussion to summarize and contextualize their main findings.

The design of Study 2 is smart and interesting. I believe it makes a larger contribution than Study 1.

Why did the authors examine text based conversations in Study 2 rather than face-to-face? How might this influence trust?

Given that the questions became more personal as the interview went on, how did the authors control for sessions that went further in the interview than others? What was the average number of questions participants completed? Might getting further in the interview be related to more closeness?

The authors note in the general discussion that the degree of deception in a conversation might be important. In Study 2, the authors report the M number of questions participants lied to. Could the authors not use this in their analyses to address this issue? Perhaps there wasn't enough variability to do so and if so the authors could just note this.

While Studies 2 and 3 were novel and interesting, I am wondering whether Study 1 is necessary/adds much to the manuscript. Perhaps the manuscript could be streamlined by excluding this study. This may also refocus the manuscript on loneliness.

Reviewer #3 (Remarks to the Author):

This manuscript provides a valuable and novel contribution to the fields of deception research and relationship science. The authors successfully explore the under-studied area of the social consequences of deception for the sender, even if their lies remain undetected. Through three studies, they demonstrate that people who lie or are otherwise untrustworthy tend to assume that others are lying and untrustworthy too, which impedes their ability to form social connections. The findings have important implications for understanding how honesty can prevent loneliness and how deception, even when undetected, can harm social connectedness.

I tend to only make suggestions or offer criticisms when I expect my suggestions may produce some substantial improvement in the quality of the manuscript. As this manuscript was quite solid, my comments will be brief and suggestions few.

The authors employed three different study designs, which strengthens the validity of their findings. The combination of a correlational study, an experimental study, and a survey of dispositional tendencies provides a solid foundation for the conclusions drawn from the research.

Though the authors consider potential biases that may have been introduced by the use of self-report measures, as well as the possibility of demand characteristics in the experimental study, a bit more on this topic would be useful in the discussion.

The sample sizes for all three studies are adequate and provide a good basis for generalization. The authors should, however, mention any possible limitations of the sample demographics and how these may affect the generalizability of the findings.

Also, the age of participants in Study 3 should be added if those data are available.

For consistency, the author may consider adding mediation figures for Study 3.

The authors should consider elaborating on practical applications of these findings, such as interventions or recommendations for individuals and organizations to promote social connectedness and interpersonal cohesion via honesty.

In conclusion, this manuscript offers a valuable contribution to deception research and relationship science, shedding light on the under-explored area of the social consequences of deception for the sender. This manuscript has the potential to make a significant impact on the field and contribute to the understanding of how honesty can influence loneliness and social connectedness. My suggestions are minor. While I encourage the authors to consider them, I support the publication of this manuscript in its current state.

Reviewer 1 Comments

1. In the section “A False Consensus of Distrust” (Ln 87-102), the current text might create the mistaken impression that the idea is original to the current paper. Markowitz and Hancock (2018; doi.org/10.1093/joc/jqy019) call this the “False Consensus Effect for Deception.” They should be cited and credited.

Our Response: We appreciate this suggestion to clarify that a false consensus of distrust is not original to the current paper and has been discussed in previous literature. We have revised the “A False Consensus of Distrust” sub-section in our Introduction to clarify that this idea has been explored in Markowitz and Hancock (2018). Specifically, we describe this work and its’ relevance to the current investigation on p. 5-6 of the revised manuscript.

2. In figures 1 and 2, the arrows from trustworthiness to closeness and perceived honesty to IOS/close (i.e., mediator to outcome) could go either direction. The closer I am to someone, the more I tend to trust them and believe them. That is, both own honesty and closeness to other could predict trust and perceived honesty and fit the results equally well. This, I think, makes the mediation argument weak.

Our Response: We agree that the nature of our study designs do not preclude the possibility of alternative—potentially bidirectional—mechanisms to support the relationship between deceiver’s distrust and social connection. In the revised manuscript, we provide mediational analyses that test this alternative model (i.e., that closeness/loneliness mediates the relationship between focal participant veracity/trustworthiness and perceived honesty/trustworthiness of others). Results are presented in footnotes on ps. 10, 18, and 24. We also identify the limitation of our designs to test mediations in the revised Discussion (see p. 28).

3. There might be an alternative explanation for the honesty effects in study two where veracity was manipulated. Participants in the lie condition might presume that their partner also might be told to lie. At the very least, the experimental instructions in the lie condition might act as a trigger from the perspective of truth-default theory.

Our Response: We quite agree. This is precisely what we predicted in Study 2 with ‘deceiver's distrust’ as a mediator in the relationship between telling lies and feeling less interpersonal closeness. The presumption that participants in the lie condition might assume that their partner also might be lying was tested using the Reysen Honesty Scale. This scale measured the extent to which an individual is perceived as honest (vs. dishonest).

4. Ln 45-46 “For example, people also admit to telling an average of two lies per day, with some individuals engaging in many more,” while this is literally true, it is misleading because the

distribution is highly skewed. Yes, some people are above average, but most people are below the average. The 2 lies were for students; non-students averaged one with a median of less than one lie every other day. Also see, doi.org/10.1111/j.1468-2958.2009.01366.x.

Our Response: We have now revised this line to more accurately reflect reported results.

5. In the McCornack & Levine (1990) study, my memory is that negative outcomes were more what the lie was about than the mere act of lying. Lying about a surprise birthday party is different than lying in order to defraud someone.

Our Response: Thank you for this suggestion. We have opted to clarify by removing the McCornack & Levine (1990) citation from line 48. Rather, we have added a separate sentence that discusses McCornack & Levine (1990) findings: that negative outcomes, such as relationship dissolution, depend on the nature of the lie.

Reviewer 2's Comments

1. Abstract: The first two sentences of the abstract seem disconnected. It seems like two separate ideas and I think the authors could more clearly integrate the ideas and purpose of the study here.

Our Response: Thank you for this suggestion; we have now updated our abstract to more clearly introduce the ideas and purpose of the study.

2. The title focusses on Loneliness but wasn't examined in Study 1 and almost feels like a secondary component in Study 2 (as it wasn't part of the main experimental manipulation). The set of studies seem to largely focus on trust and closeness primarily. The loneliness findings are interesting and important but perhaps not as central as the title suggests. Furthermore, the title suggests that feeling alone is the consequence of telling lies, however, this temporal order was not assessed. Perhaps loneliness results in greater lie-telling. Given that only correlations of loneliness were examined the authors should be cautious with their language. This is also true for the mediation analyses. Since all measures, in all studies, were taken at a single time point, the order could be revised in the model (particularly for study 3) and the same relation found. As such the authors should be cautious with causal or directional language throughout the manuscript.

Our Response: Thank you for this suggestion. In our current manuscript, we are much more deliberate with our language; we speak of 'loneliness' only when that is specifically what we measured (i.e., Study 3 and supplementary analyses in Study 2). In Studies 1 and 2, we measured closeness and are careful to use this word to describe our results in those studies. We use causal language only as it relates to social closeness, as this was the dependent variable in our only

experimental study (i.e., Study 2). In addition, we have amended the title to better reflect the set of studies—using ‘social connection’ as an umbrella term to encompass both loneliness and social closeness.

3. Social Consequences of honesty: the authors note that others have not examined the social consequences of honesty, however, Dykstra has two studies examining the longitudinal associations between lying and secret keeping to friends and to parents and the associations with relationship quality and depressive symptoms. I think these papers (both in 2020) would help strengthen the authors’ argument that lying has social consequences for the lie-teller. Furthermore, Dykstra argues that the bidirectional association between lying/secret keeping and depression is likely due to the lie-teller concealing depression but also lying isolating the lie-teller from their support system. These findings support the authors’ ideas that lying is related to loneliness and that there are consequences for the lie-teller. While Dykstra examines adolescence, these findings appear to be relevant here. I believe DePaulo also has work showing that adults rate social interactions more negatively when they lie which would be relevant here.

Our Response: We appreciate Reviewer 2 alerting us to Dykstra’s studies and have now included their work on the longitudinal association between lie telling and secret keeping and its association with relationship quality in adolescent-parent dyads (see p. 5).

4. A False Consensus of Distrust: the authors argue that one’s own dishonesty influences their perception of others and cite research around the sender’s perception of the deceiver. Another relevant citation here is Evans & Lee (2014) in which they examined adolescent truth and lie-tellers lie-detection of other’s truths and lies and found that participants’ own prior dishonesty biased their judgements. Again, I think this strengthens the authors’ arguments here.

Our Response: Thank you for your suggestion. We have now referenced Evans & Lee (2014) in the revised manuscript to further support our hypothesis generation.

5. The authors reported that participants were to talk for 25 minutes – can they report how long participants ended up talking for? Mean length and range of time the conversations lasted?

Our Response: We have now included the descriptive statistics for the conversations in Study 1 and Study 2. In Study 1, dyads conversed for 29 to 113 minutes with an average of 30.01 minutes ($SD = 7.79$). In Study 2 participant conversations ranged from 5.16 minutes to 28.25 minutes, with an average length of time 21.74 ($SD = 4.97$).

6. The authors sub heading is “Less Trustworthy Participants Felt Less Close to Conversational Partners” but the results are written such that more trustworthiness was positively related to closeness. Aligning these two would be helpful for the reader.

Our Response: As suggested by Reviewer #2, we have amended the results section to align with the sub heading “Less Trustworthy Participants Felt Less Close to Conversational Partners”.

7. Before presenting mediation analyses the authors should confirm that all variables were correlated. Also, there are limitations to these conclusions since the authors collected all data at a single time point.

Our Response: We agree that there are limitations to our mediation analyses and that the proposed mediator (i.e., deceivers’ distrust) and interpersonal closeness/loneliness may be reversed. We now provide the results of these alternative analyses and consider the limitations of our mediation analyses in the revised Discussion (p. 28).

8. It would be helpful at the end of each study if the authors provided a mini-discussion to summarize and contextualize their main findings.

Our Response: Thank you for your suggestion. We have now included short discussions after Studies 1, 2, and 3 to summarize and contextualize the main findings.

9. Why did the authors examine text based conversations in Study 2 rather than face-to-face? How might this influence trust?

Our Response: We chose to examine text-based conversations as this study was carried out during the COVID-19 pandemic, where we were limited to conducting studies in an online environment. We therefore opted to recruit online workers through Prolific and conducted chat-based conversations using ChatPlat software. We recognize that chat-based interactions between strangers is likely different than face-to-face conversations. For example, research suggests that people may be more willing to disclose personal information in an online environment where they feel relatively anonymous (e.g., Joinson, 2001). At the same time, social presence and interpersonal warmth is typically reduced in online compared to face-to-face interactions (Walther, 1996, 2001). Accordingly, we suggest—in our revised Discussion—that these findings be replicated in a face-to-face environment (see p. 27).

10. Given that the questions became more personal as the interview went on, how did the authors control for sessions that went further in the interview than others? What was the average number of questions participants completed? Might getting further in the interview be related to more closeness?

Our Response: This is a great question. We now report the average number of questions that participants completed in the interview in the manuscript. Additionally, we report an independent

samples *t*-test which suggests that the number of questions answered by liars and truth-tellers does not differ significantly (see p. 16). Although we did not add this to the manuscript, it is worth noting that we also added the number of questions answered as a covariate to the linear mixed models used to test H1 in Study 2, and found that this did not impact the reported main effect of veracity. In other words, the support for H1 remains, even when the number of questions answered is entered into the model.

11. The authors note in the general discussion that the degree of deception in a conversation might be important. In Study 2, the authors report the M number of questions participants lied to. Could the authors not use this in their analyses to address this issue? Perhaps there wasn't enough variability to do so and if so the authors could just note this.

Our Response: We have revised the manuscript for clarity and we hope it is clear that in Study 2, the reported mean for degree of deception is provided as a manipulation check. We calculated degree of deception for each participant, by averaging their responses for degree of deception of each conversation starter question. This confirmed that senders assigned to the lie condition did, in fact, engage in more deception than those in the truth condition. In the subsequent analyses, veracity was treated as a (dichotomous) fixed effect.

12. While Studies 2 and 3 were novel and interesting, I am wondering whether Study 1 is necessary/adds much to the manuscript. Perhaps the manuscript could be streamlined by excluding this study. This may also refocus the manuscript on loneliness.

Our Response: Thank you for the opportunity to clarify Study 1's addition to the manuscript. Study 1 is used as a foundational base to establish that there is a relationship between self-rated trustworthiness and feelings of closeness with a partner. Additionally, it provides support for further examining the mechanism of deceiver's distrust as it has not been explored in this specific context in previous literature. It also provides evidence for the proposed effect in a novel context (i.e., video-based conversations). Thus, while it is not without limitations, we do believe that it is additive to the manuscript.

Reviewer 3 Comments

1. Though the authors consider potential biases that may have been introduced by the use of self-report measures, as well as the possibility of demand characteristics in the experimental study, a bit more on this topic would be useful in the discussion.

Our Response: We appreciate these suggestions. We have revised our Discussion section to consider these limitations (see p. 27).

2. The sample sizes for all three studies are adequate and provide a good basis for generalization. The authors should, however, mention any possible limitations of the sample demographics and how these may affect the generalizability of the findings.

Our Response: In all three studies, a standard sample of participants were gathered from Prolific and was limited to individuals residing in the United States and 18 years of age or older. As such, the generalizability of our findings is limited to western, industrialized, and individualistic cultures. Per the reviewer's suggestion, we have included this in our discussion of the studies limitations (p. 27-28).

3. Also, the age of participants in Study 3 should be added if those data are available.

Our Response: Unfortunately, due to experimenter error, we did not gather age information in Study 3. A footnote has been added into the manuscript to explain this missing piece of demographic information.

4. For consistency, the author may consider adding mediation figures for Study 3.

Our Response: We have revised our manuscript to include mediation figures for Study 3 (see p. 24-25).

5. The authors should consider elaborating on practical applications of these findings, such as interventions or recommendations for individuals and organizations to promote social connectedness and interpersonal cohesion via honesty.

Our Response: Thank you for this suggestion. We have now included a few sentences in the conclusion section of our manuscript posing some potential, practical applications of our findings.

20th Jul 23

Dear Dr ten Brinke,

Your manuscript titled "On Deception and Social Connection" has now been seen by our reviewers, whose comments appear below. In light of their advice I am delighted to say that we are happy, in principle, to publish a suitably revised version in Communications Psychology under the open access CC BY license (Creative Commons Attribution v4.0 International License).

We therefore invite you to revise your paper one last time to address the remaining concerns of our reviewers and a list of editorial requests. At the same time we ask that you edit your manuscript to comply with our format requirements and to maximise the accessibility and therefore the impact of your work.

EDITORIAL REQUESTS:

You will see that we ask you to revise the Abstract in a way that makes it compliant with our guidelines and responds to the remaining referee concern.

Further, we strongly recommend that you use more visual display items to present the data underlying your analyses. Some detailed recommendations are included in the Editorial Request Table.

SUBMISSION INFORMATION:

OPEN ACCESS:

Communications Psychology is a fully open access journal. Articles are made freely accessible on publication under a [CC BY](http://creativecommons.org/licenses/by/4.0) license (Creative Commons Attribution 4.0 International License). This license allows maximum dissemination and re-use of open access materials and is preferred by many research funding bodies.

For further information about article processing charges, open access funding, and advice and support from Nature Research, please visit a

[href="https://www.nature.com/commpsychol/article-processing-charges">https://www.nature.com/commpsychol/article-processing-charges](https://www.nature.com/commpsychol/article-processing-charges)

At acceptance, you will be provided with instructions for completing this CC BY license on behalf of all authors. This grants us the necessary permissions to publish your paper. Additionally, you will be asked to declare that all required third party permissions have been obtained, and to provide billing information in order to pay the article-processing charge (APC).

* **CODE AVAILABILITY:** All Communications Psychology manuscripts must include a section titled "Code Availability" at the end of the methods section. We require that the custom analysis code supporting your conclusions is made available in a publicly accessible repository at this stage; this requirement pertains for example to SPSS and R based analyses, even if you did not create novel software. please choose a repository that generates a digital object identifier (DOI) for the code; the link to the repository and the DOI must be included in the Code Availability statement. Publication as Supplementary Information will not suffice.

* **DATA AVAILABILITY:**

[link redacted]

Best regards,

Marike

Marike Schiffer, PhD
Chief Editor
Communications Psychology

REVIEWERS' COMMENTS:

Reviewer #1 (Remarks to the Author):

The authors generally did a good job addressing my previous comments.

I only have one minor quibble with the revisions:

I do not believe that the first 6 words in the abstract are true (or at least they are misleading). I would be happy to debate the evidence with the authors, but I think Serota et al.'s program of research is clear this claim has been debunked.

The revised claim in the first paragraph on the same point is literally accurate, but now clearly stated. The point is that the average is inflated.

Reviewer #2 (Remarks to the Author):

I think the authors have done a wonderful job addressing reviewers comments and feedback and have no additional comments/suggestions.

Reviewer #3 (Remarks to the Author):

I believe that all of the reviewers' concerns have been adequately addressed and the manuscript should be accepted.